# Diagnosis and Stratification of COVID-19 Infections Using Differential Plasma Levels of D-Dimer: A Two-Center Study from Saudi Arabia

Abdullah Alsrhani [1],*, Ahmad Alshomar [2], Abozer Y Elderdery [1], Zafar Rasheed [3] and Aisha Farhana [1]

[1]   Department of Clinical Laboratory Sciences, College of Applied Medical Sciences, Jouf University, Sakaka 72388, Saudi Arabia
[2]   Department of Medicine, College of Medicine, Qassim University, Buraidah 51452, Saudi Arabia
[3]   Department of Medical Biochemistry, College of Medicine, Qassim University, Buraidah 51452, Saudi Arabia
*   Correspondence: afalserhani@ju.edu.sa

**Abstract:** Background: D-dimer, generated upon the degradation of fibrin, is extensively used to detect thrombosis in various diseases. It is also explored as a marker for thrombosis in cases with COVID-19 disease. Few studies have confirmed its utility as a marker for assessing disease severity. Objectives: The current research was undertaken to determine the role of D-dimer in patients with COVID-19 and to investigate any association with the progression and severity of the disease in the Saudi population. Methods: Clinical indices in confirmed COVID-19 patients were collected from tertiary care hospitals in Aljouf and Qassim regions. The plasma D-dimer levels were quantified directly in the samples collected from COVID-19 patients ($n = 148$) using an immunofluorescence assay, and the data were presented in Fibrinogen Equivalent Units (mg/L). The collected data of D-dimer were analyzed based on COVID-19 severity, age, and the gender of patients. Results: The findings show that the plasma D-dimer concentrations were significantly ($p = 0.0027$) elevated in COVID-19 cases ($n = 148$), compared to in the normal healthy uninfected controls ($n = 309$). Moreover, the D-dimer levels were analyzed according to the severity of the disease in the patients. The data revealed that D-dimer concentrations were significantly increased in patients with mild infection to moderate disease, and the levels were the highest in patients with severe COVID-19 disease ($p < 0.05$). Our analysis demonstrates that the D-dimer levels have no association with the age or gender of COVID-19 patients ($p > 0.05$) in the study population. Conclusions: D-dimer can serve as a biomarker not only for the detection of COVID-19 infection, but also for determining the severity of infection of COVID-19 disease.

**Keywords:** COVID-19; D-dimer; disease severity; age; gender; biomarker



## 1. Background

The identification of the first case of COVID-19 in 2019 was followed by its worldwide spread, and it was finally recognized as a pandemic by the World Health Organization in March 2020 [1]. It took a global toll with many deaths [2]. Although the pandemic wave is presently trailing, its global health, social, and economic impact has been immense and continues to pose a threat with its mutant forms. Most COVID-19 patients are asymptomatic, while others show varied symptoms, such fever, coughing, breathlessness, and mild pneumonia [3]. Some cases may develop severe pneumonia, followed by hypoxia and respiratory dysfunction [4]. An average of 5% of symptomatic patients also exhibit acute respiratory distress syndrome (ARDS), which can progress to multiple organ failure [4,5]. Severe illness and death primarily result from the development of acute ARDS, sepsis, and multiple organ failure, stemming from dysfunctional immunological, endothelial, and coagulation responses, triggered by viral infection. Among the abnormal laboratory

findings, thrombocytopenia, leukopenia, the development of a hypercoagulative state, and raised D-dimer concentrations, often require intensive care unit admissions [5,6].

Although the evaluation of biochemical and hematological indices, in order to understand the infectivity and severity of the disease, remains a promising approach for COVID-19 diagnosis, recently, the genetic make-up of the host has been envisioned as a critical determinant [7]. A striking relationship is observed between COVID-19 disease and distinct host gene polymorphisms, supported by the observed variability in the spread and severity of the disease across different ethnicities [8]. COVID-19 susceptibility, severity, and disease outcome reflect an association with gene expression patterns, mutations, deletions, and polymorphisms [8,9]. Some genetic polymorphisms in host ACE1, APOE, and IFITM3 genes demonstrate an increased infection risk, while polymorphisms in ACE2, AGTR1, TMPRSS2, VDR, and TNFA are associated with the severity of disease [10–17]. Studies have also indicated a high infectivity potential and increased severity of COVID-19 in the A blood group serotype, while the O blood group serotype is observed to be protective against infection [18]. Genetic polymorphism, SNPs, in-host immune response genes, including interferons, TNF, and interleukins, are attributed to inducing a high COVID-19 severity [19]. Additionally, besides the genetic and immune determinants, age and gender association has been widely observed in COVID-19 infectivity and severity [20,21]. Gender variation in COVID-19 remains multifactorial, and is mainly affected by lifestyle. However, it may also be attributed to the genetic differences in the key disease modifiers; for example, ACE2 regulation by estrogen explains its reduced infectivity and severity potential [21,22]. Existing morbid conditions are closely linked to a higher susceptibility and severity of COVID-19 [23–25].

Many studies have documented an elevation in D-dimer and plasma fibrinogen concentrations in the initial stages of COVID-19 infection [6,26]. Generally, D-dimer is one of the parameters used to detect thrombosis in several pathological conditions [27,28]. They are formed as a product when plasmin breaks down blood clots by cleaving fibrin. A three to four-fold increase in plasma D-dimer is related to a poor disease prognosis [28]. An elevation in D-dimer levels is also noted in COVID-19 cases, reflecting underlying conditions, such as cancer, diabetes, stroke, and pregnancy [29–32].

Monitoring the levels of D-dimer and other coagulation indices from the initial stages of the disease has shown promise in its management [33]. Worldwide, several studies have confirmed increased D-dimer in severe COVID-19 cases, reaching a high value in critically ill patients and a high death rate [33,34]. The COVID-19-related coagulopathy is presumed responsible for the elevation of D-dimer. Meta-analysis studies have also demonstrated the association of venous thromboembolism with COVID-19 severity [5]. A correlation between the D-dimer and COVID-19 disease, and age and gender-related disease severity, has been recognized [35,36]. Sporadic reports have also shown an association between elevated D-dimer levels, and the delta and omicron variants [37].

Incidences of infection by COVID-19 variants continue to trickle through, even when the global COVID-19 wave has subsided. Urgency is still present if the infection is detected in a patient with underlying conditions, such as cardiovascular disease or diabetes mellitus. Management of such cases can be improved by monitoring the D-dimer, which can predict outcomes [29,30].

Present research on the diagnostic importance of D-dimer in assessing COVID-19 severity is encouraging, yet it is still in the potential phase and mandates confirmation [38–40]. Nonetheless, D-dimer holds promise as a prognostic biomarker, alongside procalcitonin, C-reactive protein, and ferritin, for assessing COVID-19 severity. Hence, the difference in the plasma levels of D-dimer can be used to detect underlying comorbidities and co-infections in patients with COVID-19 [15,41,42]. This will help to facilitate clinical management that is both more personalized and more efficient, allowing speedier recovery and significantly reducing the mortality rate.

## 2. Objectives

In our study, we aimed to measure the D-dimer plasma concentration of COVID-19 patients, in order to ascertain any correlation with disease severity, age, and gender.

## 3. Methods

### 3.1. Study Design and Clinical Setting

This study included 148 patients diagnosed with COVID-19 infection between January and November 2021, attending hospitals in Aljouf and Qassim Regions in Saudi Arabia. Patient enrollment was limited to hospital encounters, including inpatient and emergency, to ascertain the availability of demographic data and clinical results of all confirmed cases. The demographic and biochemical parameters were collected and analyzed (Table 1). The D-dimer concentration was compared between patients with severe and non-severe infection, and between age and gender.

**Table 1.** Summary of COVID-19 patients.

| Parameters | COVID-19 Patients (*n* = 148) | Normal Range |
|---|---|---|
| Age (years) Mean ± SD (range) | 58.7 ± 17.7 (19.0 to 95.0) | - |
| Sex, M/F | 60/38 | - |
| Body mass index (BMI) Mean ± SD (range) | 31.7 ± 9.1 (15 to 81) | 18.5 to 24.9 |
| White Blood Cell (leukocytes), $10^3/\mu L$ Mean ± SD (range) | 7.95 ± 4.1 (1.8 to 23.0) | 4.0 to 11.0 |
| Absolute neutrophils count, $10^3/\mu L$ Mean ± SD (range) | 5.97 ± 3.9 (0.57 to 21.0) | 2.0 to 6.9 |
| Absolute lymphocyte count, $10^3/\mu L$ Mean ± SD (range) | 1.25 ± 0.6 (0.20 to 4.30) | 0.6 to 4.0 |
| Platelet count, $10^3/\mu L$ Mean ± SD (range) | 235.4 ± 84.9 (91.0 to 532.0) | 150.0 to 450.0 |
| Creatinine, $\mu mol/L$ Mean ± SD (range) | 88.9 ± 62.1 (24.0 to 531.0) | 64.0 to 104.0 |
| Alanine Transaminase (ALT), IU/L Mean ± SD (range) | 48.7 ± 91.1 (6.0 to 487.0) | 5.0 to 55.0 |
| Total Bilirubin, $\mu mol/L$ Mean ± SD (range) | 10.7 ± 5.7 (0.60 to 27.0) | 3.4 to 20.5 |
| D-dimer, mg/L FEU) Mean ± SD (range) | 1.35 ± 1.7 (0.05 to 6.1) | 0.00 to 0.50 |

Patients with any secondary infection, including bacterial, viral, and fungal, or with insufficient pretreatment clinical results, were excluded. A criterion was followed to confirm COVID-19 positivity, comprising at least two (+ve) findings of the RT-PCR (Real-Time Reverse Transcriptase-Polymerase Chain Reaction) assay. Patients who showed negative COVID-19 test results were also excluded. Healthy infected subjects, who had never had COVID-19 disease and were confirmed negative by a COVID-19 test before sampling, were included as a control. The severity of the disease was assessed by $SpO_2$ <94%, biochemical tests, and chest X-rays.

### 3.2. Laboratory Testing

The clinical parameters of the confirmed positive cases of COVID-19 were obtained from the hospitals, and blood samples from the same patients collected into blue-top sodium citrate tubes were used to assess biochemical parameters. The D-dimer was quantified

directly in the plasma of positive COVID-19 (*n* = 148) samples using an automated, standardized immunofluorescence assay, and the data were presented in Fibrinogen Equivalent Units (FE mg/L). The immunofluorescence assay for D-dimer testing is well established, a highly standardized procedure in our lab routinely used for testing D-dimer in many pathological conditions. The data for D-dimer were analyzed based on the severity of COVID-19, age, and gender of the patients. The normal level for the D-dimer assay used was 0.23 mg/L FEU, equivalent to 230 ng/mL, set as an upper limit.

### 3.3. Statistical Analysis

Statistical analysis was carried out using GraphPad Prism-9 (GraphPad Software Inc., San Diego, CA, USA) to determine the prevalence, mean, and 95% CI. A comparison analysis was carried out using one-way or two-way ANOVA. A *p*-value of < 0.05 and lower was considered statistically significant.

## 4. Results

### 4.1. D-Dimer Can Differentiate between Positive and Negative COVID-19 Cases

We observed that the concentration of D-dimer in COVID-19 patients (*n* = 148) was significantly high (1.37 ± 1.71 mg/L FEU), compared to non-infected healthy (*n* = 309) controls (0.37 ± 1.62 mg/L FEU), with a *p*-value of 0.0027. Figure 1 indicates that the difference in D-dimer values between non-infected healthy subjects (NHS) and COVID-19 patients is very significant.

### 4.2. D-Dimer Can Be Used for COVID-19 Disease Stratification

The D-dimer levels were also analyzed regarding the hospitalization status of the COVID-19 patients. The results show that the D-dimer levels were significantly increased between the patient groups with mild infection to moderate infection, and were highest in patients with severe COVID-19 disease (*p* < 0.05).

### 4.3. Variations in D-Dimer Concentration Do Not Corroborate with the Age or Gender of COVID-19 Infected Patients

Analysis of the data to ascertain gender variations (male and female) and age ranges did not show any significant differences in our study sample (*p* > 0.05).

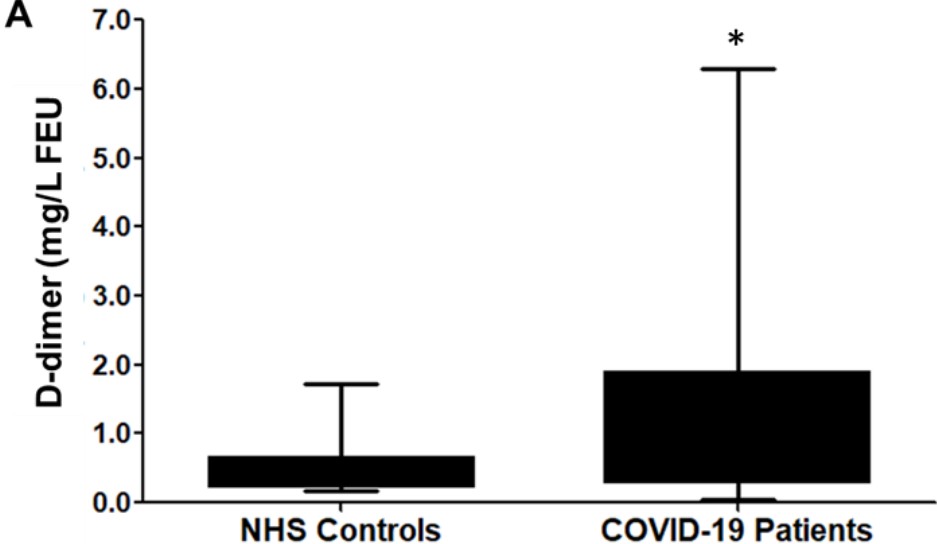

**Figure 1.** *Cont.*

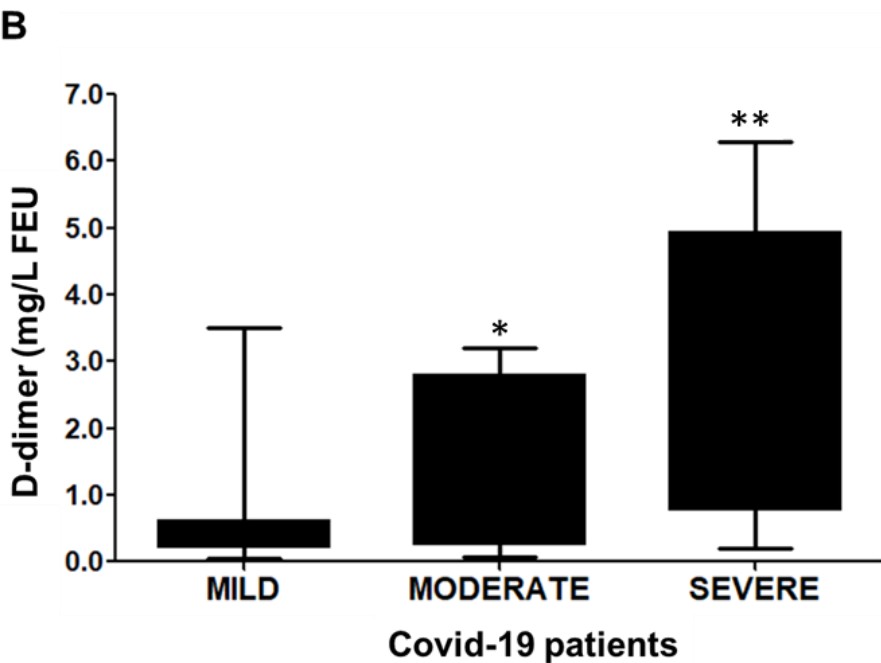

**Figure 1.** D-dimer in COVID-19 patients. The graphs represent the D-dimer concentration in mg/L FEU± SD. (**A**) D-dimer levels in the blood samples of COVID-19 patients (*n* = 148) and in never-infected human subjects (NHS, *n* = 309 A * *p* = 0.0027 were obtained in COVID-19 vs. NHS. A comparison analysis was carried out using the Mann–Whitney test. (**B**) D-dimer levels in COVID-19 patients with mild (*n* = 66), moderate (*n* = 48), and severe (*n* = 34) infection were highly significant with a, * *p* < 0.05 (95% CI −2.285 to −0.04205) vs. mild infection; ** *p* < 0.05 (95% CI −2.591 to −0.1751) vs. moderate infection; and ** *p* < 0.001 (95% CI −3.243 to −1.850) vs. severe infection. Box and Whisker show the Min to Max values with median ±SD. Comparison analysis was performed via the one-way ANOVA method, followed by Tukey's post hoc test.

## 5. Discussion

With moderate heterogeneity, the diagnostic potential of D-dimer has shown promise in assessing the COVID-19 disease progression. Similar to other studies, we also observed a significant (*p*-value of ≤ 0.001) increase in the plasma D-dimer concentrations in COVID-19 patients (1.37 ± 1.71 mg/L FEU), compared to healthy controls never infected with COVID-19 (0.37 ± 1.62 mg/L FEU) (Figure 1A). Furthermore, we also observed a significant association between D-dimer levels and the severity of the disease, indicating a sequential elevation in the D-dimer concentration with an increase in the severity of the disease (Figure 1B). The comparison of the plasma D-dimer values, among mild (0.54 ± 1.77 mg/L FEU, moderate (1.88 ± 0.74 mg/L FEU) and severe (3.23 ± 1.2 mg/L FEU) COVID-19 disease, point to a very significant (*p* ≤ 0.05) difference in plasma D-dimer concentrations; this reflects its use as a diagnostic index. The D-dimer concentrations for mild, moderate, and severe cases corroborated with the clinically-confirmed disease severity, evaluated through low $SpO_2$ values, biochemical tests, and radiographs for COVID-19 patients.

COVID-19 disease management depends on the stratification of the disease pathogenesis and fostering treatment for the patients accordingly [43,44]. In this regard, a diagnostic value in evaluating the severity of the disease may facilitate better disease management. Increased levels of D-dimer among COVID-19 cases also reflect inflammatory responses to other infections, plus endothelial cell dysfunction leading to thrombin production and hypoxic conditions [45,46]. D-dimer is observed to be persistently higher than normal in about 15% of patients who have experienced severe COVID-19 disease, indicating an underlying pathology as an effect of COVID-19 [47]. Thus, a distinctive value of D-dimer, associated with different pathological conditions in COVID-19 disease, may help to evaluate and stratify COVID-19 cases according to the severity of the disease.

There exists a differential risk for thrombosis in males and females, and quite a few studies have indicated a relationship between gender and age with the D-dimer values in COVID-19 patients [48,49]. This difference in the risk of the thrombotic pattern may be attributed to the difference in genetic make-up and gene expression patterns in males vs. females. The role of host-genetic variation is currently being tested as an underlying cause of COVID-19 susceptibility and severity [50]. Recently, cohort studies and meta-analyses have observed a difference in the elevation of D-dimer among males and females. They propose a corroboration to be mandated through further studies to achieve a distinct conclusion [51]. Although there is a gender bias in the outcome of COVID-19 infection in Saudi Arabia, our results do not show any marked difference ($p > 0.05$) in the D-dimer values between males and females (Figure 2). The effect of gender on D-dimer values may be more clearly discernible if the sample size is increased. We also assessed age-dependent variation in the D-dimer levels by categorizing the COVID-19 patients into above 50 years and below 50 years groups, but no significant ($p > 0.05$) association was observed (Figure 3). The D-dimer correlation with age and COVID-19 disease has been variable in the literature. Some studies indicate a severe disease pattern in males, with no correlation to age [52]. On the other hand, some studies observe a correlation between D-dimer and disease severity with age [53,54]. In our study, we do not observe D-dimer differences in relation to the age and gender of the patient (Figures 2 and 3). Hence, based on the results of our study, D-dimer does not appear to be a confounding factor in assessing COVID-19 severity according to age.

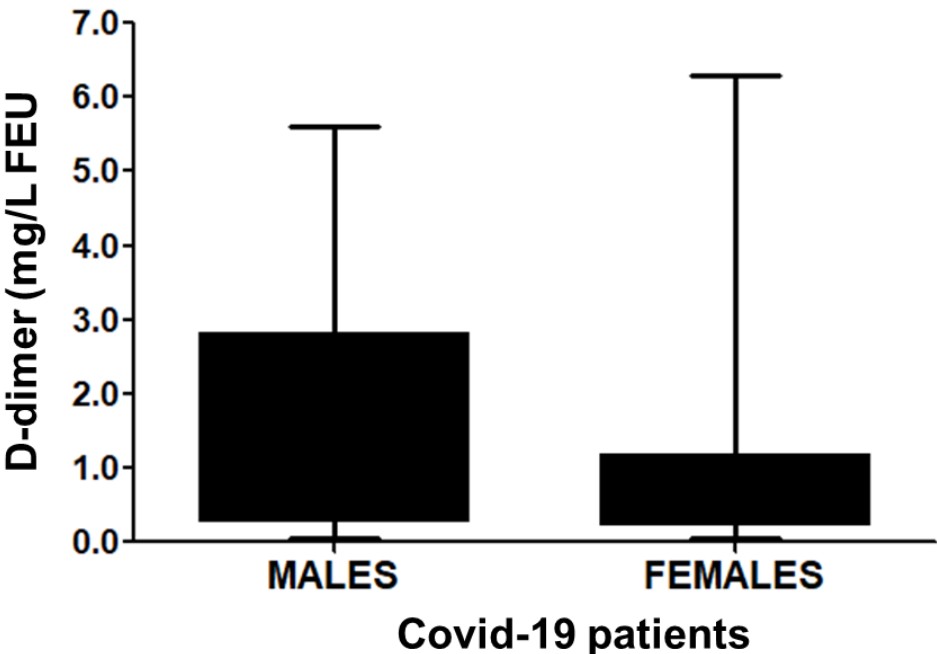

**Figure 2.** D-dimer levels in male and female COVID-19 patients. Of the COVID-19 patients, levels of D-dimer in the blood samples of males (*n* = 97) and females (*n* = 51) were measured. Male COVID-19 patients vs. female COVID-19 patients, *p* = 0.4488 (95% CI -0.4277 to 0.9575). Box and Whisker show the Min to Max values with median ± SD.

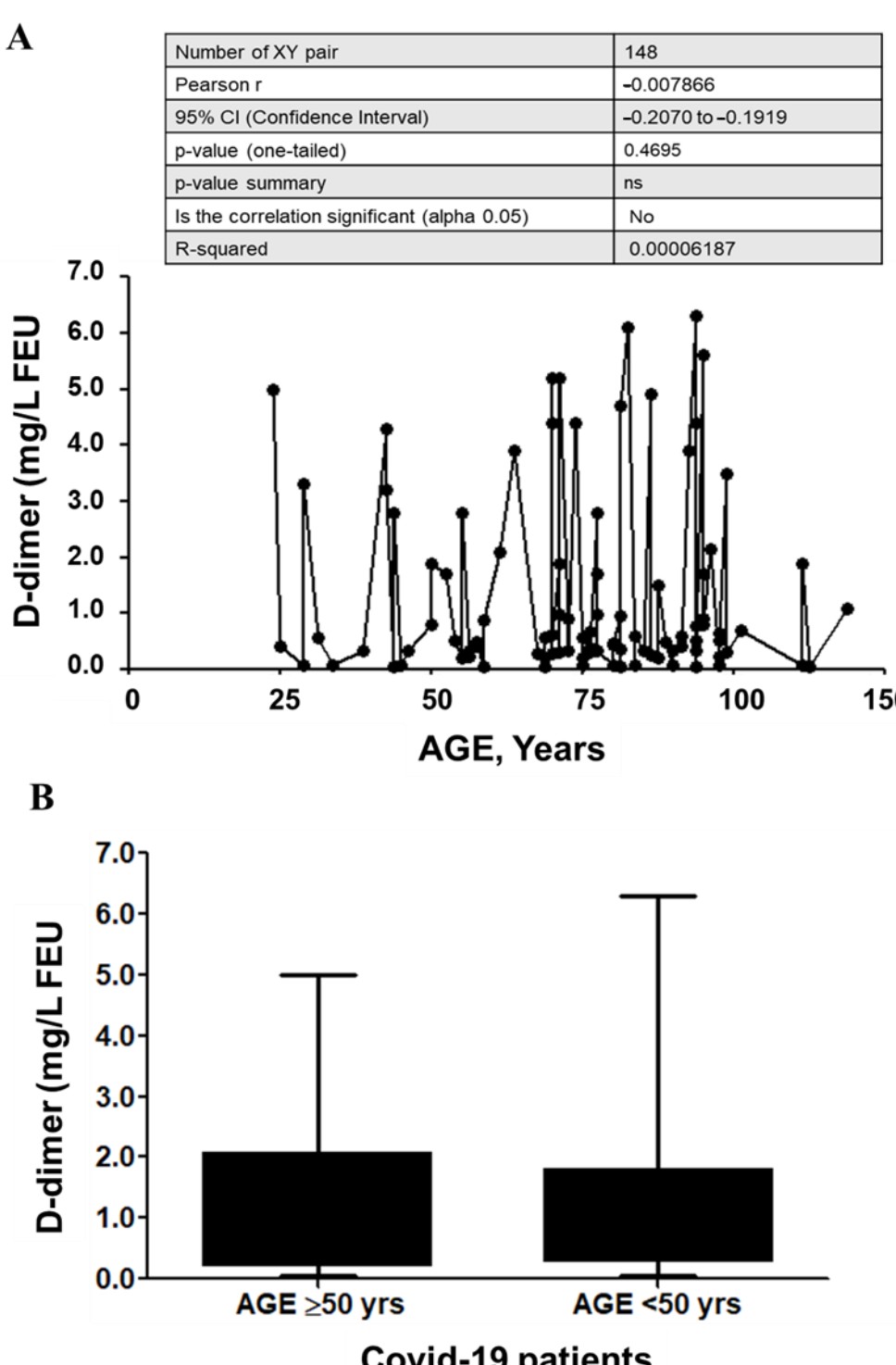

**Figure 3.** Age-wise distribution of D-dimer levels in COVID-19 patients. (**A**) D-dimer correlation with the age of COVID-19 patients. Pearson r = −0.007866; *p* = 0.4695 (95% CI −0.2070 to 0.1919). (**B**) D-dimer levels in COVID-19 cases with age ≥50 years (Age ≥ 50 years; *n* = 41) and age < 50 years (Age < 50 years; *n* = 107), *p* = 0.6869. Box and Whisker show the Min to Max values with median ± SD. Comparison analysis was carried out using a two-way ANOVA method and the Bonferroni post hoc test.

## 6. Conclusions

Our study aims to investigate the role played by D-dimer in COVID-19 patients, evaluating any association with the progress and severity of the disease. Our findings

indicate that high D-dimer levels are specifically related to COVID-19 progression. In patients with symptoms that progress to pulmonary complications, the utility of D-dimer as a potential biomarker, used to monitor COVID-19 severity, is relevant to stratify the cases and recommend a specific treatment regime. Furthermore, a high or increasing D-dimer level can provide prognostic information, useful for assessing COVID-19 patients at risk of developing severe disease. This study concludes that the concentration of D-dimers can serve as a valuable biomarker to stratify COVID-19 disease patients according to severity, and to diagnose the presence of a pro-thrombotic state.

**Author Contributions:** Conceptualization: A.A. (Abdullah Alsrhani), Z.R. and A.F.; Methodology: Z.R. and A.F.; Investigation, Z.R. and A.A. (Ahmad Alshomar); Writing original draft: A.F. and A.A. (Abdullah Alsrhani); Analysis: A.F. and Z.R.; Review and editing, A.A. (Abdullah Alsrhani) and A.Y.E. All authors have read and agreed to the published version of the manuscript.

**Funding:** The authors thank the Deanship of Scientific Research, Jouf University, for funding this research, through grant number 40/297.

**Institutional Review Board Statement:** The study was conducted in accordance with the Declaration of Helsinki, and approved by the research ethics committee of Qurayyat Health Affairs (protocol code 109 and 30 December 2021 of approval).

**Informed Consent Statement:** Not applicable.

**Data Availability Statement:** The data generated during the study is contained within the article.

**Acknowledgments:** The authors thankfully acknowledge the hospital staff in King Abdul Aziz Specialty Hospital, Aljouf region, Saudi Arabia for the assistance and support during this research project.

**Conflicts of Interest:** The authors declare no conflict of interest.

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
