# Peer review of "Diagnosis and Stratification of COVID-19 Infections Using Differential Plasma Levels of D-Dimer: A Two-Center Study from Saudi Arabia"

_2036-7481, doi:10.3390/microbiolres14010006_

Round 1

Reviewer 1 Report

In the paper “Plasma Levels of D-Dimer as a Potential Biomarker for Prediction and Stratification of Severity in Covid-19 Infections”, the authors address one of the fundamental issues in the fight against COVID-19 pandemic: the identification of predictive biomarkers for the severity of the disease.

Overall, the paper is well written, and the topic has a good scientific relevance. Anyway, some issues must be addressed.

D-dimer has already been identified as a potential biomarker and a paper entitled “D-Dimer as a potential biomarker for disease severity in COVID-19” has been published. This paper (doi: 10.1016/j.ajem.2020.12.023) is not mentioned by the authors, and it should be. Also, the title should be modified to highlight the point that the present work strengthens previously identified data and to be more different from the previous one.

In my opinion, the introduction should be widened to contain a broader discussion about the different factors that can modify the response to SARS-CoV-2. A wide literature has been published in the last two years, especially regarding the role of polymorphisms, that could be usefully introduced here. I hereby suggest some reviews to begin:

·       10.1016/j.gene.2022.146790

·       10.1016/j.htct.2021.07.006

·       10.1016/j.gene.2022.146674

·       10.1002/jmv.27849

·       10.1016/j.ejmg.2021.104227

·       10.17179/excli2022-4976

·       10.1002/jmv.27615

·       10.3906/biy-2104-67

·       10.3390/genes11091010

Materials and Methods should clarify where the “non-infected healthy” samples come from: are they people that never had COVID-19 or did they experience the disease previously? If so, are the related data available?

Please, revise the symbols used for the figures: the hash (#) is not commonly used for significancy. Please, use a different number of asterisks (*) for different significances.

Typos are spread through the paper, please check it.

Author Response

Point-by-point reply to reviewer’s comments:

REVIEWER 1:

In the paper “Plasma Levels of D-Dimer as a Potential Biomarker for Prediction and Stratification of Severity in Covid-19 Infections”, the authors address one of the fundamental issues in the fight against COVID-19 pandemic: the identification of predictive biomarkers for the severity of the disease. Overall, the paper is well written, and the topic has a good scientific relevance. Anyway, some issues must be addressed.

Reply: We thank the reviewer for the insightful comments that helped to improve the quality of our work.

D-dimer has already been identified as a potential biomarker and a paper entitled “D-Dimer as a potential biomarker for disease severity in COVID-19” has been published. This paper (doi: 10.1016/j.ajem.2020.12.023) is not mentioned by the authors, and it should be. Also, the title should be modified to highlight the point that the present work strengthens previously identified data and to be more different from the previous one.

Reply: Thank you for the comment. Accordingly, the changes have been incorporated in the title and in the background section (line 68-69).

In my opinion, the introduction should be widened to contain a broader discussion about the different factors that can modify the response to SARS-CoV-2. A wide literature has been published in the last two years, especially regarding the role of polymorphisms,  that could be usefully introduced here. I hereby suggest some reviews to begin:

  • 10.1016/j.gene.2022.146790
  • 10.1016/j.htct.2021.07.006
  • 10.1016/j.gene.2022.146674
  • 10.1002/jmv.27849
  • 10.1016/j.ejmg.2021.104227
  • 10.17179/excli2022-4976
  • 10.1002/jmv.27615
  • 10.3906/biy-2104-67
  • 10.3390/genes11091010

Reply: Thank you for the comment which has helped to strengthen the results in reflecting additional host factors for the stratification of covid-19 disease. The above studies and additional related researches are now added in the introduction (48-67).

Materials and Methods should clarify where the “non-infected healthy” samples come from: are they people that never had COVID-19 or did they experience the disease previously? If so, are the related data available?

Reply: We have accordingly modified the sentence in material and method section 3.1 (line 114-116).

Please, revise the symbols used for the figures: the hash (#) is not commonly used for significancy. Please, use a different number of asterisks (*) for different significances.

Reply: The # symbol has now been changed to * and ** representing comparison in Figure 1A and 1B.

Typos are spread through the paper, please check it

Reply: Typo errors have been corrected in the revised manuscript.

Reviewer 2 Report

1.       The authors should discuss the reason for opting the immunofluorescence based method for plasma D-dimer quantification in comparison to the classical enzyme-linked immunosorbent assays (ELISA), and latex agglutination assay methods.

2.       Figure 1, 2 and 3B, please re-confirm if these large error bars are SD or SEM. I feel let the data speak itself, by using dot plot rather than box and whiskers.

3.       In the present study the authors have looked for clinical indices of 148 Covid-19 positive patients and found no sex differences in D-dimer concentrations. In a recent retrospective cohort study involving 4574 patients, it was reported that males with COVID-19 had greater prognostic value for D-dimer (https://doi.org/10.1016/j.hrtlng.2022.10.012). A similar study involving 11827 hospitalized COVID-19+ adults too showed that male sex had higher mean D-dimer plasma levels and were at higher risk of experiencing poor COVID-19 outcomes than the females (https://abstracts.isth.org/abstract/the-impact-of-sex-on-d-dimer-levels-and-disease-outcomes-in-hospitalized-covid-19-patients/). Can the authors justify this observed discrepancy?  

4.       The authors should also discuss the fact that persistently elevated D-dimers levels are observed even in Covid-19 recovered patients too (doi:10.1371/journal.pone.0258351).

5.       In the present study, the authors have not found any corroboration between D-dimer levels and age of Covid-19 infected patients and went on to conclude that D-dimer levels have no association with the age and gender of Covid-19 patients (p >0.05). This finding doesn’t correlate with the current literature (doi: 10.3390/jcm11123298; doi: 10.3390/jcm10225433) and hence needs further explanations.

Author Response

Point-by-point reply to reviewer’s comments:

REVIEWER 2:

We thank the reviewer for the valuable comments on the technical and logical aspect of the paper that has helped to make our research better.

1.The authors should discuss the reason for opting the immunofluorescence based method for plasma D-dimer quantification in comparison to the classical enzyme-linked immunosorbent assays (ELISA), and latex agglutination assay methods.

Thank you for the comment. We have used immunofluorescence based assay because it is a highly standardized technique in our lab for D-dimer analysis. We routinely use this technique to assess D-dimer in various pathological samples that includes samples from cardiac disease, and other diseases. The technique is well established in the lab with unlikely chance of error in the results. A sentence to clarify the use of immunofluorescence assay instead of classical methods has now been added in the paper (line 124-126).

  1. Figure 1, 2 and 3B, please re-confirm if these large error bars are SD or SEM.

Reply: The error bars represent SD values. The same has now been included in the legends of Figures 1, 2 and 3B (line 123, 165 and170.

  1. In the present study the authors have looked for clinical indices of 148 Covid-19 positive patients and found no sex differences in D-dimer concentrations. In a recent retrospective cohort study involving 4574 patients, it was reported that males with COVID-19 had greater prognostic value for D-dimer (https://doi.org/10.1016/j.hrtlng.2022.10.012). A similar study involving 11827 hospitalized COVID-19+ adults too showed that male sex had higher mean D-dimer plasma levels and were at higher risk of experiencing poor COVID-19 outcomes than the females (https://abstracts.isth.org/abstract/the-impact-of-sex-on-d-dimer-levels-and-disease-outcomes-in-hospitalized-covid-19-patients/). Can the authors justify this observed discrepancy?

Reply: Thank you for the insightful comment. Indeed many studies have observed a difference in D-dimer among males and females. We have also mentioned this point in discussion section (line 204-206). The above mentioned study involved a distinctively high number of subject to score for the outcome of the disease in relation to the severity pattern, demonstrating the odds of Covid19 severity in males with elevation in troponin, irrespective of subsequent D-dimer elevation. The study indicated that D-dimer may be a prognostic marker proposing that studies on covid-19 and thrombosis to confirm its sex dependent variation.

The meta-analysis study also indicates a significant difference in the male and female D-dimer values. However, the results are limited by the difference in D-dimer reporting values across studies (as mentioned in the paper).

In our paper, we only propose a diagnostic value of D-dimer wherein an increase in D-dimer possibly suggest the level of severity and is not related to the outcome of the disease. However, as advised, we have now further discussed it in the discussion section (line 207-216)

  1. The authors should also discuss the fact that persistently elevated D-dimers levels are observed even in Covid-19 recovered patients too (doi:10.1371/journal.pone.0258351).

Reply: We have now included and discussed the above paper (line 202-204).

  1. In the present study, the authors have not found any corroboration between D-dimer levels and age of Covid-19 infected patients and went on to conclude that D-dimer levels have no association with the age and gender of Covid-19 patients (p >0.05). This finding doesn’t correlate with the current literature (doi: 10.3390/jcm11123298; doi: 10.3390/jcm10225433) and hence needs further explanations.

Reply: We have removed the sentence in line 165-167. We have also discussed the result in relevance with other studies (line 218-223).
